# Enhancing Maize Transformation and Targeted Mutagenesis through the Assistance of Non-Integrating *Wus2* Vector

**DOI:** 10.3390/plants12152799

**Published:** 2023-07-28

**Authors:** Minjeong Kang, Keunsub Lee, Qing Ji, Sehiza Grosic, Kan Wang

**Affiliations:** 1Department of Agronomy, Iowa State University, Ames, IA 50011, USA; mjkang@iastate.edu (M.K.); klee@iastate.edu (K.L.); 2Crop Bioengineering Center, Iowa State University, Ames, IA 50011, USA; jjessica@iastate.edu (Q.J.); shadzic@iastate.edu (S.G.); 3Interdepartmental Plant Biology Major, Iowa State University, Ames, IA 50011, USA

**Keywords:** *wuschel2*, morphogenic transcription factor, *Agrobacterium*-mediated transformation, ternary vector system, CRISPR-Cas9, transgene-free editing

## Abstract

Efficient genetic transformation is a prerequisite for rapid gene functional analyses and crop trait improvements. We recently demonstrated that new T-DNA binary vectors with *NptII*/G418 selection and a compatible helper plasmid can efficiently transform maize inbred B104 using our rapid *Agrobacterium*-mediated transformation method. In this work, we implemented the non-integrating *Wuschel2* (*Wus2*) T-DNA vector method for *Agrobacterium*-mediated B104 transformation and tested its potential for recalcitrant inbred B73 transformation and gene editing. The non-integrating *Wus2* (NIW) T-DNA vector-assisted transformation method uses two *Agrobacterium* strains: one carrying a gene-of-interest (GOI) construct and the other providing an NIW construct. To monitor *Wus2* co-integration into the maize genome, we combined the maize *Wus2* expression cassette driven by a strong constitutive promoter with a new visible marker *RUBY*, which produces the purple pigment betalain. As a GOI construct, we used a previously tested CRISPR-Cas9 construct pKL2359 for *Glossy2* gene mutagenesis. When both GOI and NIW constructs were delivered by LBA4404Thy- strain, B104 transformation frequency was significantly enhanced by about two-fold (10% vs. 18.8%). Importantly, we were able to transform a recalcitrant inbred B73 using the NIW-assisted transformation method and obtained three transgene-free edited plants by omitting the selection agent G418. These results suggest that NIW-assisted transformation can improve maize B104 transformation frequency and provide a novel option for CRISPR technology for transgene-free genome editing.

## 1. Introduction

Recent advances in genome-editing tools, i.e., CRISPR technology, have demonstrated a great potential for modern crop breeding through *de novo* domestication [1,2,3] as well as precise trait engineering [4]. In particular, CRISPR-Cas9 has been widely used for different applications including, but not limited to, targeted gene mutagenesis via indel mutations, targeted DNA insertions, precise base changes, RNA editing, transcriptional reprogramming, and chromosomal restructuring [5,6,7,8]. CRISPR technology or any other genome-editing tools, however, cannot be fully utilized without establishing efficient methods for reagent delivery and the regeneration of fertile plants, i.e., genetic transformation [9].

Maize (*Zea mays* L.) is one of the most important cereal crops worldwide, but efficient genetic transformation methods are only available for a limited number of genotypes or inbreds in the public sector [10]. We recently established a rapid maize inbred B104 transformation method, which only takes about two months to produce rooted transgenic plants [11]. Our ternary vector system, which consists of a T-DNA binary vector and a compatible helper plasmid, demonstrated a robust B104 transformation efficiency with a different reporter and CRISPR-Cas9 vectors [11,12]. In particular, the adoption of the *neomycin phosphotransferase II* (*NptII*)/G418 selection system drastically reduced the escape rate from 36–67% to less than 10%, greatly saving time and resources compared to the *bar*/bialaphos selection system [12]. In this study, we implemented a non-integrating *ZmWus2* vector-assisted transformation method [13] to further enhance B104 transformation frequency and tested its potential for recalcitrant genotype B73 transformation and gene editing.

Morphogenic transcription factors (MTF), such as *Wuschel2* (*Wus2*) and *Baby Boom* (*Bbm*), are widely applicable to enhance transformation frequency and expand amenable genotypes [14,15,16]. However, constitutive expression of *Wus2* or *Bbm* prevents normal plant regeneration and production of fertile plants [14], thus requiring the removal of the *Wus2*/*Bbm* expression cassettes by *Cre*-mediated inducible excision [17]. To circumvent this issue, Corteva Agriscience’s scientists demonstrated an alternative approach using an additional *Agrobacterium* strain harboring a T-DNA binary vector that carries a *Wus2* expression cassette with a strong constitutive promoter [13] (Figure 1a). This approach is based on the observation that Wus2 proteins are diffusible to neighboring cells and can stimulate cell proliferation without the integration of *Wus2* T-DNA [14,18]. Using an NIW vector, we tested three thymidine–auxotrophic *Agrobacterium* strains for maize B104 immature embryo transformation. In addition, we tested if transgene-free gene editing can be achievable for a recalcitrant inbred B73 using the NIW-assisted transformation method. We observed significantly improved B104 transformation frequency and obtained three transgene-free gene-edited B73 plants, suggesting that the NIW-assisted transformation method can be a useful option for maize genome engineering.

## 2. Results

### 2.1. NIW-Assisted Transformation of Maize B104

We made an NIW vector pKL2391 (Figure 1b) carrying a strong constitutive promoter for maize *Wus2* expression and a visible reporter *RUBY*, which converts tyrosine to purple pigment betalain [19], to monitor the integration of the NIW construct T-DNA. A CRISPR-Cas9 vector pKL2359 (Figure 1c) [12] was used as a GOI construct and delivered by LBA4404Thy- strain in all experiments. The NIW construct pKL2391 was delivered by one of the three thymidine–auxotrophic *Agrobacterium* strains for B104 immature embryo transformation. *Agrobacterium* cells were prepared as described by Hoerster et al. [13]. The GOI and NIW strains were mixed at a 9:1 ratio immediately before the immature embryo infection [13].

As summarized in Table 1, NIW-assisted transformation using *Agrobacterium* strain LBA4404Thy- (Thy-, *thymidine synthase* knockout mutant) [20] significantly enhanced B104 transformation frequency compared to the control (18.8% vs. 10.0%; *p* < 0.05, two proportion *z*-test). NIW strain EHA105TR (TR, *thymidine synthase,* and *recA* knockout) [21] also enhanced transformation frequency by 55% (15.5% vs. 10.0%; *p* = 0.055, two proportion *z*-test) but it was not statistically significant. One strain, EHA105Thy- [22], did not have a noticeable impact on the average transformation frequency compared to the control (6.8% vs. 10.0%; *p* = 0.241, two proportion *z*-test). Our previous study using LBA4404Thy- strain harboring pKL2359 had a similar result with 7.7% of transformation frequency, which is comparable to the control in this study [12], suggesting that NIW-assisted transformation can be utilized to further enhance maize transformation efficiencies.

RUBY reporter was very useful to identify *Wus2*-integrated callus or regenerated plantlets. Betalain accumulation could be easily identified, and callus pieces or regenerating shoots with purple color were excluded during the subculture, resulting in a low *Wus2* co-integration frequency (Table 1; 2/176 = 1.1%). As previously demonstrated [12], the CRISPR-Cas9 showed efficient targeted mutagenesis, and most of the analyzed T0 plants (80.8–89.5%) carried various indel mutations at the *Glossy2* (*Gl2*) target site (Appendix A).

### 2.2. Transgene-Free Gene Editing in Maize B73

Maize inbred B73 is an important public genotype that serves as a reference genome and has been widely utilized in many research programs [23]. However, genetic transformation or gene editing of B73 remains challenging due to its recalcitrance to regeneration. We wanted to explore the potential of the NIW-assisted transformation approach for transgene-free gene editing in B73. Transgene-free CRISPR-Cas9-mediated gene editing has been demonstrated before using a transient expression system in wheat [24]. Recalcitrant genotypes such as B73, however, pose an additional challenge: i.e., poor regeneration efficiency. We tested if the NIW-assisted transformation approach can be combined with the transient expression of CRISPR-Cas9 to generate gene-edited B73 plants without transgene integration (Figure 2a). We omitted selection agent G418 from all tissue culture media to recover both transgenic and transgene-free regenerants (Figure 2b).

As summarized in Table 2, we did not recover any regenerated plants from the 136 infected immature embryos with our rapid transformation method (control), confirming the “recalcitrant” nature of the B73 inbred. On the other hand, the NIW-assisted transformation approach was efficient in stimulating B73 regeneration as we recovered 79 independent plants from 395 infected immature embryos (Table 2). Betalain pigmentation was not observed from the 79 regenerants, and PCR results confirmed that *Wus2* T-DNA co-integration was not detected from the regenerated plants. We used PCR screening to identify transgenic plants and nine were positive for the CRISPR-Cas9 T-DNA (9/79 = 11.4%), indicating that the vast majority were transgene-free regenerants (70/79 = 88.6%; Table 2). We then PCR-amplified the CRISPR-Cas9 target, *Gl2* exon2 region [11,12], and used Sanger sequencing and TIDE/ICE analyses [25,26] to identify gene-edited plants. We found a total of 10 plants that carried indel mutations at the target site: seven T0 plants (Appendix A) and three transgene-free plants (Table 3). Seven T0 plants showed biallelic (BI), heterozygous (HT), or mosaic (MO) mutation at the target site (1 HT, 2 BI, and 4 MO plants; Appendix A). A loss-of-function mutant was identified among the regenerated T0 plants and showed the expected phenotype (water droplet adherence to the surface of young leaves) when misted (Figure 2c). Importantly, three *gl2*-edited plants were transgene-free (Appendix A), indicating that the NIW-assisted transformation approach can be adopted for transgene-free gene editing applications for recalcitrant genotypes such as B73 (Table 3). In our case, one plant had a biallelic mutation (−4/−27), and the other two had heterozygous mutations (+1/0; −1/0), demonstrating an efficient targeted mutagenesis without CRISPR-Cas9 T-DNA integration (Table 3).

## 3. Discussion

In this study, we implemented a non-integrating *Wus2* approach [13,27] for NIW-assisted transformation of maize inbred lines B104 and B73. When combined with our rapid B104 transformation method [11], overall transformation frequency was enhanced by about two-fold (10% vs. 18.8%) by simply mixing two *Agrobacterium* strains that carry GOI and NIW constructs, respectively, with a 9:1 ratio before immature embryo infection. This simple extra step can significantly save time and resources for researchers to produce transgenic or gene-edited plants. Because the NIW *Agrobacterium* strain is separately maintained, only GOI *Agrobacterium* strains need to be modified for different applications, making the NIW-assisted transformation method a versatile option for enhancing maize transformation efficiency.

We tested three different *Agrobacterium* strains for their capacity to deliver the NIW T-DNA. One strain LBA4404Thy- significantly enhanced B104 transformation frequency, whereas the other two strains, EHA105Thy- and EHA105TR, did not have a marked impact (Table 1). Because *Wus2* co-integration inhibits normal plant regeneration, each NIW *Agrobacterium* strain might have a different optimal mixing ratio to effectively deliver NIW T-DNA without co-integration with the GOI T-DNAs. In addition, because EHA105Thy- and EHA105TR have a different chromosomal background (*A. tumefaciens* C58) to LBA4404Thy- (*A. tumefaciens* Ach5), there might be different cell-to-cell interactions, i.e., interbacterial competition, when mixed for co-infection. Future studies might provide further insights into how different *Agrobacterium* strains can perform in combination with other strains to deliver NIW and GOI T-DNAs.

As a proof of concept, we applied the NIW-assisted transformation approach for a recalcitrant genotype B73 to generate transgene-free gene-edited plants. Importantly, NIW-assisted transformation was efficient in stimulating B73 regeneration without *Wus2* integration, as we recovered 79 regenerants from 395 infected embryos (Table 2). In contrast, none were regenerated from the 136 infected embryos of control. Molecular analyses identified nine transgenic plants and three transgene-free edited plants. Based on the number of infected immature embryos, these numbers can be translated to 2.3% of transformation frequency for the transgenic plants (9/395) and 0.8% of the transgene-free targeted mutagenesis rate (3/395). In addition, based on the number of regenerated plants, 11.4% were transgenic and 3.8% were transgene-free edited plants. It needs to be noted that among the three transgene-free edited plants, one was a biallelic mutant and the others were heterozygous mutants; therefore, loss-of-function edited plants without transgenes can be readily obtained in the first generations as biallelic or homozygous mutants, or they can be obtained in the second generation as homozygous mutants after segregation. Given that our rapid transformation method only takes two months to produce first-generation edited plants, transgene-free edited plants can be obtained in about 2–6 months for phenotyping analyses. Interestingly, previous studies reported that an *Agrobacterium* strain carrying a mutant *virD2* gene exhibited significantly impaired T-DNA integration frequency while the transient T-DNA expression was only slightly diminished [28,29]. It would be worthwhile to develop engineered *Agrobacterium* strains that can efficiently deliver T-DNAs into plant cells without integration. MTFs and CRISPR reagents might be delivered as a single T-DNA to achieve transgene-free gene editing.

## 4. Materials and Methods

### 4.1. NIW Vector Construction

The NIW binary vector was constructed using pTF101.1 [30] as a backbone. First, pTF101.1 DNA was digested with restriction enzymes *Hind*III and *Aat*II and the 7.4 kb fragment was purified after agarose gel electrophoresis. Next, a 2.6 kb fragment containing 3× viral enhancers and maize ubiquitin promoter and a 1.5 kb fragment containing maize *Wus2* coding sequence and *In2-1* gene terminator were PCR-amplified from PHP97334 [31], which was kindly provided by Dr William Gordon-Kamm at Corteva Agriscience, using primers listed in Appendix A (ZmWUS2-F1 and TIN2-1-R1; 3×ENH-F1 and PZmUbi-R1). All PCR reactions were carried out using high-fidelity Q5 DNA polymerase (NEB, Ipswich, MA, USA). HiFi DNA assembly mix (NEB) was used for the Gibson assembly, and the resulting plasmid was named pKL2387. This intermediate vector was digested with *Kpn*I and dephosphorylated using alkaline phosphatase to prevent self-ligation. The *RUBY* reporter with CaMV 35S promoter [19] was PCR-amplified from pCBL101-RUBY [12] using two primer pairs (P35-RUBY-F1 and DODA-R1; DODA-F1 and P35-RUBY-R1) listed in Appendix A. Gibson assembly using the NEB Hifi DNA assembly mix resulted in the NIW binary vector pKL2391 (Figure 1b).

### 4.2. Agrobacterium Transformation

Three thymidine–auxotrophic *Agrobacterium* strains were tested for NIW-assisted transformation: LBA4404Thy- [20], EHA105Thy- [22], and EHA105TR [21]. LBA4404Thy- strain was used for the GOI construct pKL2359 (Figure 1c, Addgene#199721; [12]) in all experiments. The electroporation method was used to introduce a ternary helper plasmid pKL2299 (Addgene #186332; [11]) and one of the T-DNA binary vectors, pKL2391 or pKL2359 into *Agrobacterium* cells [32]. Transformed *Agrobacterium* cells were spread on YEP medium (10 g/L Yeast extract, 5 g/L NaCl, 10 g/L Bacto peptone, 15 g/L Bacto agar) amended with thymidine (50 mg/L) and appropriate antibiotics (100 mg/L of Gentamicin and 100 mg/L of Spectinomycin for pKL2391; 100 mg/L of Gentamicin and 50 mg/L of Kanamycin for pKL2359).

### 4.3. NIW-Assisted Transformation of Maize B104 and B73

Seeds of the maize inbred lines were obtained from Dr Erik Vollbrecht (Iowa State University), and immature embryos were produced in the Crop Bioengineering Laboratory greenhouse, Iowa State University (Ames, IA, USA). Maize plants were grown as previously described [11]. Briefly, the greenhouse conditions were as follows: photoperiod, 14 h/10 h (day/night); temperatures, 28 °C/22 °C (day/night). All silks were covered with shoot bags before the emergence to prevent cross-contamination. Each ear was self-pollinated or crossed with sibling plants. B104 and B73 ears containing embryos of appropriate size (1.8–2.0 mm) were harvested 12–14 days after pollination.

*Agrobacterium* suspension for infection was prepared as described by Kang et al. (2022) [11]. Briefly, a mother plate was prepared by streaking *Agrobacterium* cells from a glycerol stock on an AB solid media containing 50 mg/L of thymidine and appropriate antibiotics and incubating at 28 °C for 2 days. A working plate was streaked on a YEP medium amended with appropriate antibiotics and thymidine using the cells from the mother plate a day before the infection. Mother plates can be stored at 4 °C for up to 10 days for streaking working plates. *Agrobacterium* suspension for infection was prepared after embryo isolation. Freshly harvested cells from a working plate were thoroughly resuspended by vortexing in 700A medium supplemented with thymidine (50 mg/L) and acetosyringone (100 µM). The final cell density was adjusted to 0.45–0.55 at OD_550nm_. GOI and NIW *Agrobacterium* suspensions were prepared separately, and two *Agrobacterium* strains were mixed with a 9:1 ratio (GOI: NIW) immediately before embryo infection.

Dehusked maize ears were disinfected using 20% bleach solution (5.25% sodium hypochlorite) for 20 min and rinsed three times with sterile water. The top of the kernels was removed using a sharp scalpel and embryos were carefully isolated using a sterile micro-spatula. Isolated embryos were collected in 2 mL tubes containing 700A medium and washed once with 700A medium before infection. To infect the embryos, the 700A medium in the 2 mL tubes was carefully removed by pipetting and 1 mL of *Agrobacterium* suspension was added. After 5 min, the embryos were poured onto a co-cultivation medium, and the excessive amount of infection medium was removed by gentle pipetting. Infected embryos were carefully re-orientated with the scutellum side up and incubated at 20 °C overnight (16–20 h) in the dark. After co-cultivation, all embryos were transferred to a resting medium (scutellum side up) and cultured for one week at 28 °C in the dark. Transient expression of *mCherry* could be checked 72 h after infection. Developing tissue with somatic embryos was transferred to a maturation medium containing 75 mg/L of G418 and cultured for 4–6 weeks at 28 °C in the dark. To regenerate transgene-free B73 plants, the selection agent G418 was omitted in all tissue culture media. Tissues on the maturation medium were sub-cultured every 2 weeks. Actively regenerating tissues with immature shoots were further transferred to a rooting medium amended with G418 and kept in a light chamber for 2–3 weeks at 28 °C under a 16h/8h photoperiod (light/dark). Rooted plantlets were carefully transferred to 32-cell seed-growing trays filled with autoclaved potting mix Metromix 360 (Sungro, Agawam, MA, USA).

### 4.4. Genotyping of Regenerated Plants

Total genomic DNA was isolated from regenerated plants using a modified version of the protocol described by Edwards et al. (1991) [33]. PCR screening was conducted to identify transgenic T0 plants using primers specific to the zCas9 gene (zCas9-F and zCas9-R; Appendix A). To identify transgene-free B73 regenerants, a total of seven primer pairs were used to check the presence of *zCas9*, *mCherry*, *NptII*, and *RUBY* genes (Appendix A). Targeted mutagenesis of the *Glossy2* gene was analyzed by Sanger sequencing and TIDE/ICE analyses [25,26] as previously described [11,34]. When the genotyping results by TIDE and ICE analyses were not clear, PCR products were cloned into pJET1.2 vector as instructed by the manufacturer (Thermo Fisher Scientific, Waltham, MA, USA), and 8–12 clones were sequenced per sample using the sequencing primers provided by the kit (pJET1.2 Forward and Reverse sequencing primers).

## 5. Conclusions

We adopted the non-integrating maize *Wus2*-assisted *Agrobacterium*-mediated transformation method to enhance B104 transformation efficiency and to produce transgene-free gene-edited plants for the recalcitrant genotype B73. When LBA4404Thy- strain was used for both NIW and GOI constructs, B104 transformation frequency was almost doubled by simply mixing a GOI *Agrobacterium* strain with an NIW strain at a 9:1 ratio, suggesting that this method can be widely applied for maize transformation and gene editing. In conclusion, our results suggest that when combined with the rapid *Agrobacterium*-mediated transformation method, NIW-assisted transformation can boost B104 transformation efficiency and provide a novel option for transgene-free gene editing for recalcitrant genotypes such as B73.

## Figures and Tables

**Figure 1 plants-12-02799-f001:**
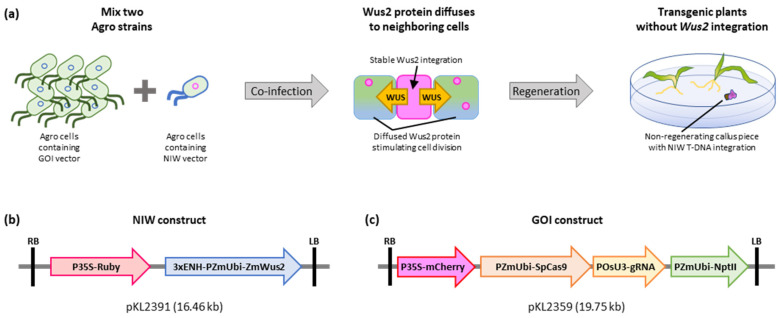
NIW-assisted transformation. (**a**) Schematic illustration of NIW-assisted maize transformation. *Agrobacterium* cells harboring GOI and NIW constructs are mixed at a 9:1 ratio before immature embryo infection, and Wus2 protein diffuses to the neighboring cells through the plasmodesmata and stimulates cell proliferation, promoting the regeneration of cells without *Wus2* integration. (**b**) T-DNA region of NIW construct pKL2391. (**c**) T-DNA region of GOI construct pKL2359. LB and RB, left and right T-DNA border sequences; P35S-*RUBY*, *RUBY* reporter consisting of betalain biosynthesis genes (*CYP76AD1*, *DODA*, and *Glucosyl transferase*) driven by CaMV 35S promoter and *Arabidopsis heat shock protein 18.2* gene terminator; 3×ENH-PzmUbi-*ZmWUS2*, maize *Wus2* expression cassette driven by 3× viral enhancers, maize ubiquitin promoter, and *In2-1* gene terminator; P35S-*mCherry*, a red fluorescent protein (*mCherry*) expression cassette driven by CaMV 35S promoter and soybean vegetative storage protein terminator; PzmUbi-*SpCas9*, maize codon optimized *Cas9* from *Streptococcus pyogenes* (*SpCas9*) with maize ubiquitin promoter and *rbcS-E9* gene terminator; PosU3-gRNA, a single-guide RNA targeting maize *Glossy2* gene driven by rice U3 promoter; PzmUbi-*NptII*, *neomycin phosphotransferase II* (*NptII*) gene driven by maize ubiquitin promoter and *potato proteinase inhibitor II* gene terminator.

**Figure 2 plants-12-02799-f002:**
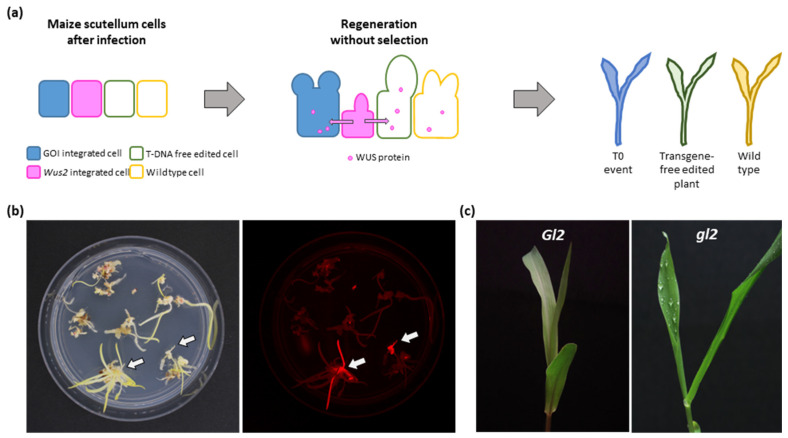
Transgene-free B73 gene editing using NIW-assisted transformation method. (**a**) Schematic illustration of B73 gene editing using NIW-assisted transformation method. Diffused Wus2 protein promotes B73 regeneration without NIW integration and both transgene-free and T-DNA-integrated plants can be regenerated by omitting the selection agent G418. (**b**) Regenerating B73 shoots, 35 days post-infection (**left**, bright field; **right**, RFP channel). Arrows indicate *mCherry*-expressing transgenic immature shoots. (**c**) Loss-of-function mutant phenotype by *Glossy2* gene knockout. Water droplets adhere to the leaf surface of the mutant plant when misted.

**Table 1 plants-12-02799-t001:** Summary of NIW-assisted transformation of B104 with different *Agrobacterium* strains.

	#Emb	#Reg Plants	#T0	TF *	#Co-in	#Wus2	#Esc
Control	420	45	42	10.0% a	0	0	3
LBA4404Thy-	384	76	72	18.8% b	1	0	3
EHA105Thy-	439	31	30	6.8% a	0	0	1
EHA105TR	477	85	74	15.5% ab	1	0	10

#Emb, number of infected immature embryos; #Reg plants, number of regenerated plants; #T0, number of transgenic plants with pKL2359 integration only; TF, transformation frequency (#T0 per 100 Emb); #Co-in, number of transgenic plants with pKL2391 and pKL2359 T-DNA co-integration; #Wus2, number of transgenic plants with pKL2391 integration only; #Esc, number of escapes (non-transgenic); Control, infection without NIW *Agrobacterium* strain; LBA4404Thy-, auxotrophic LBA4404 strain with *thymidine synthase* gene (*thyA*) knockout; EHA105Thy-, auxotrophic EHA105 strain with *thyA* knockout; EHA105TR, *recA*-deficient auxotrophic EHA105Thy- strain. * TFs denoted with the same letter are not significantly different at *p* < 0.05 (two proportion *z*-test).

**Table 2 plants-12-02799-t002:** Summary of non-selective NIW-assisted transformation of B73.

	#Emb	#Reg Plants	#T0	TF	#Co-in	#Wus2	#WT
Control	136	0	0	0.0%	0	0	0
LBA4404Thy-	395	79	9	2.3%	0	0	70

#Emb, number of infected immature embryos; #Reg plants, number of regenerated plants; #T0, number of transgenic plants with pKL2359 T-DNA integration only; TF, transformation frequency (#T0 per 100 Emb); #Co-in, number of transgenic plants with pKL2391 and pKL2359 T-DNA co-integration; #Wus2, number of transgenic plants with pKL2391 T-DNA integration only; #WT, number of wild type regenerants (non-transgenic); Control, infection without NIW *Agrobacterium* strain; LBA4404Thy-, co-infection with NIW *Agrobacterium* strain.

**Table 3 plants-12-02799-t003:** Transgene-free editing of *Glossy2* gene in B73.

Plant ID.	T0 Genotype		*Glossy2* Sequence	Indel Mutation	Contribution %
	**WT**	Allele 1:	TTGGTCACAGATCACAAACTTCAAATGCGGTGGGCTGGCGCTGGGGTTCAGCT	0 bp	
		Allele 2:	TTGGTCACAGATCACAAACTTCAAATGCGGTGGGCTGGCGCTGGGGTTCAGCT	0 bp	
**B73-R1**	**BI**	Allele 1:	TTGGTCACAGATCACAAACT----ATGCGGTGGGCTGGCGCTGGGGTTCAGCT	−4 bp	67
		Allele 2:	TTGGTCACAGATCACAAACT---------------------------TCAGCT	−27 bp	33
**B73-R2**	**HT**	Allele 1:	TTGGTCACAGATCACAAACTTCAAAATGCGGTGGGCTGGCGCTGGGGTTCAGCT	+1 bp	38
		Allele 2:	TTGGTCACAGATCACAAACTTCAAATGCGGTGGGCTGGCGCTGGGGTTCAGCT	0 bp	38
**B73-R3**	**HT**	Allele 1:	TTGGTCACAGATCACAAACTTCA-ATGCGGTGGGCTGGCGCTGGGGTTCAGCT	−1 bp	40
		Allele 2:	TTGGTCACAGATCACAAACTTCAAATGCGGTGGGCTGGCGCTGGGGTTCAGCT	0 bp	42

WT, wild type; BI, biallelic mutation; HT, heterozygous mutation. PAM (blue) and protospacer (red) sequences are highlighted. Inserted base (A) is underlined. Contribution % represents relative proportions of indel sequence (*p* < 0.001) in each sample.

## Data Availability

All study data that support the findings are included in the article or Appendix A.

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
