# Peer review of "Enhancing Maize Transformation and Targeted Mutagenesis through the Assistance of Non-Integrating Wus2 Vector"

_plants, 2023, doi:10.3390/plants12152799_

Round 1
Reviewer 1 Report
The manuscript by Kang et al. established the Agrobacterium-mediated transformation platform using a non-integrating Wuschel2 T-DNA vector (NIW) to transform a recalcitrant inbred maize variety B104 for gene editing experiments. The NIW-assisted transformation method uses two Agrobacterium strains, one carrying CRISPR reagents and the other with the NIW construct. Recently, some reports suggested that fine-tuning Wuschel2, morphogenic transcription factors, could enhance crop transformation frequency. Here authors have tested such strategies for their experiments. Additionally, they inserted a visible marker gene RUBY into the NIW cassette that produces purple pigment betalain for visual identification of transformed cells. They designed CRISPR-guide RNA to target the phenotypic marker gene Glossy2 as a proof of concept. Finally, they claimed that NIW constructs could significantly enhance the transformation efficiency in maize varieties.
The experimental methodology is correct and presented in detail. However, some significant aspects need to be addressed.
Major comments are as follows
The authors tabulated the transformation efficiency data, and I can suggest incorporating the genome editing efficiency data too in that table.
NIW approach increases the escape rate (Table S1), and the authors want to make any insight into that as a limitation.
Authors performed the ICE analyses for genotyping, here (table 3) they should incorporate the allelic abundance (%).
Line 191-193 “one was biallelic mutant and the others were heterozygous mutants, suggesting that loss-of-function edited plants without transgenes can be readily obtained in the first or second generations”. They only analysed CRISPR-Cas9 edited first second generations (T0) plants hence such a claim is overvalued.
Author Response
The manuscript by Kang et al. established the Agrobacterium-mediated transformation platform using a non-integrating Wuschel2 T-DNA vector (NIW) to transform a recalcitrant inbred maize variety B104 for gene editing experiments. The NIW-assisted transformation method uses two Agrobacterium strains, one carrying CRISPR reagents and the other with the NIW construct. Recently, some reports suggested that fine-tuning Wuschel2, morphogenic transcription factors, could enhance crop transformation frequency. Here authors have tested such strategies for their experiments. Additionally, they inserted a visible marker gene RUBY into the NIW cassette that produces purple pigment betalain for visual identification of transformed cells. They designed CRISPR-guide RNA to target the phenotypic marker gene Glossy2 as a proof of concept. Finally, they claimed that NIW constructs could significantly enhance the transformation efficiency in maize varieties.
The experimental methodology is correct and presented in detail. However, some significant aspects need to be addressed.
Major comments are as follows
(Q1) The authors tabulated the transformation efficiency data, and I can suggest incorporating the genome editing efficiency data too in that table.
Our response: We respectively disagree. Adding the genome editing data into Table 1 will make it difficult to read the data. Thus we would like to separate the transformation efficiency and editing frequency data in Table 1 and Table S1, respectively.
(Q2) NIW approach increases the escape rate (Table S1), and the authors want to make any insight into that as a limitation.
Our response: We have not observed any significant increase in the escape rate by the NIW approach, as summarized in Table 1 (last column). The escape rate for the control was 6.7% (3/45), while those for the NIW approach ranged from 3.2-11.8% with an average of 7.3% (14/192).
(Q3) Authors performed the ICE analyses for genotyping, here (table 3) they should incorporate the allelic abundance (%).
Our response: Thanks for the suggestion. We incorporated the allelic abundance data in the last column of Tables 3 and S3.
(Q4) Line 191-193 “one was biallelic mutant and the others were heterozygous mutants, suggesting that loss-of-function edited plants without transgenes can be readily obtained in the first or second generations”. They only analysed CRISPR-Cas9 edited first second generations (T0) plants hence such a claim is overvalued.
Our response: Thanks for the suggestion. We have revised the sentence for improved clarity. Because we obtained one biallelic (B73-R1) and two heterozygous mutants (B73-R2 and B73-R3) without transgene in the first generation, we reasoned that loss-of-function mutants could be readily obtained in the first (i.e., biallelic or homozygous mutant) or second generation (i.e., homozygous mutant by inheritance).
Before revision:
“It needs to be noted that among the three transgene-free edited plants, one was biallelic mutant and the others were heterozygous mutants, suggesting that loss-of-function edited plants without transgenes can be readily obtained in the first or second generations.”
After revision:
“It needs to be noted that among the three transgene-free edited plants, one was a biallelic mutant, and the others were heterozygous mutants; therefore, loss-of-function edited plants without transgenes can be readily obtained in the first generation as biallelic or homozygous mutants or they can be obtained in the second generation as homozygous mutants after segregation.”
Reviewer 2 Report
It is necessary to give a brief description of the plant material. The method of cultivation, to which variety of corn the inbred B104 belongs.
The authors note different chromosomal background and differences in intercellular interactions, please describe this in more detail.
What positive qualities do the authors think will prevail in bivalent and heterozygous mutants obtained as a result of research?
The list of references should be brought in line with the requirements of the journal.
Author Response
(Q1) It is necessary to give a brief description of the plant material. The method of cultivation, to which variety of corn the inbred B104 belongs.
Our response: Thanks for the suggestion. The B104 is yellow dent corn which was developed as part of the Iowa Stiff Stalk Synthetic breeding program in 1996. Due to its genetic background, it shows high sequence similarity with the reference maize line B73 (Manchanda et al., 2016). In the methods part, we cited a protocol paper from our group ‘Kang et al., 2022’ which describes plant material preparation with great details. We have added a couple of sentences in the main text about greenhouse care.
Before revision:
Maize plants were grown as previously described [11]. B104 and B73 ears containing embryos of appropriate size (1.8 – 2.0 mm) were harvested 12-14 days after pollination.
After revision:
Maize plants were grown as previously described [11]. Briefly, the greenhouse conditions were as follows: photoperiod, 14 h/10 h (day/night); temperatures, 28°C/22°C (day/night). All silks were covered with shoot bags prior to the emergence to prevent cross-contamination. Each ear was self-pollinated or crossed with sibling plants. B104 and B73 ears containing embryos of appropriate size (1.8 – 2.0 mm) were harvested 12-14 days after pollination.
(Q2) The authors note different chromosomal background and differences in intercellular interactions, please describe this in more detail.
Our response: Thanks for the suggestion. Agrobacterium strains LBA4404Thy- and EHA105Thy- have different chromosomal backgrounds: LBA4404Thy- was derived from Ach5, whereas EHA105Thy- has C58 chromosomal background with disarmed Ti plasmid from Bo542. Different chromosomal backgrounds might trigger interbacterial competition among the mixed Agrobacterium strains, which could affect Agrobacterium-mediated T-DNA delivery efficiency. We have added this information to the text.
Before revision:
In addition, because EHA105Thy- and EHA105TR have a different chromosomal background (A. tumefaciens C58) to LBA4404Thy- (A. tumefaciens Ach5), there might be different cell-to-cell interactions when mixed for co-infection.
After revision:
In addition, because EHA105Thy- and EHA105TR have a different chromosomal background (A. tumefaciens C58) to LBA4404Thy- (A. tumefaciens Ach5), there might be different cell-to-cell interactions, i.e., interbacterial competition, when mixed for co-infection.
(Q3) What positive qualities do the authors think will prevail in bivalent and heterozygous mutants obtained as a result of research?
Our response: We assume that the reviewer was referring to the transgene-free edited plants from B73. The fact that we obtained three transgene-free edited plants with either biallelic or heterozygous mutations suggests that the NIW approach can be useful for genome editing applications without transgene integration. The persistence of transgenes or CRISPR reagents in the plant genome after editing can cause regulatory concerns and undesired side effects such as random insertional mutagenesis or off-target effects. Because NIW-assisted transient expression of CRISPR reagents can avoid such negative impacts while shortening the time to obtain transgene-free edited plants, we think that our NIW-assisted approach can be widely applicable for precise genome editing studies.
(Q4) The list of references should be brought in line with the requirements of the journal.
Our response: The list of references has been revised as suggested. The journal name has been updated to the abbreviated name.
References
Manchanda, N., Andorf, C. M., Ye, L., Wimalanathan, K., Rounsley, S., Wang, K., & Lawrence-Dill, C. (2016, March). Sequencing, assembly, and annotation of maize B104: a maize transformation resource. In Meeting Abstract (p. 92).
Kang, M., Lee, K., Finley, T., Chappell, H., Veena, V., & Wang, K. (2022). An improved Agrobacterium-mediated transformation and genome-editing method for maize inbred B104 using a ternary vector system and immature embryos. Frontiers in Plant Science, 13, 860971.